# 3D Bioprinting of Hydrogels for Cartilage Tissue Engineering

**DOI:** 10.3390/gels7030144

**Published:** 2021-09-16

**Authors:** Jianghong Huang, Jianyi Xiong, Daping Wang, Jun Zhang, Lei Yang, Shuqing Sun, Yujie Liang

**Affiliations:** 1Department of Orthopedics, Shenzhen Second People’s Hospital (Health Science Center, First Affiliated Hospital of Shenzhen University), Shenzhen 518035, China; huangjh20@mails.tsinghua.edu.cn (J.H.); jianyixiong@126.com (J.X.); wangdp@mail.sustech.edu.cn (D.W.); yiyuanbgs@126.com (L.Y.); 2Tsinghua University Shenzhen International Graduate School, Shenzhen 518055, China; zhangjun@sticmail.sz.gov.cn; 3Institute of Biomedicine and Health Engineering, Tsinghua University Shenzhen International Graduate School, Shenzhen 518055, China; 4Department of Child and Adolescent Psychiatry, Shenzhen Kangning Hospital, Shenzhen Mental Health Center, Shenzhen 518020, China

**Keywords:** additive manufacturing, 3D bioprinting, bioinks, hydrogel, cartilage tissue engineering

## Abstract

Three-dimensional (3D) bioprinting is an emerging technology based on 3D digital imaging technology and multi-level continuous printing. The precise positioning of biological materials, seed cells, and biological factors, known as “additive biomanufacturing”, can provide personalized therapy strategies in regenerative medicine. Over the last two decades, 3D bioprinting hydrogels have significantly advanced the field of cartilage and bone tissue engineering. This article reviews the development of 3D bioprinting and its application in cartilage tissue engineering, followed by a discussion of the current challenges and prospects for 3D bioprinting. This review presents foundational information on the future optimization of the design and manufacturing process of 3D additive biomanufacturing.

## 1. Introduction

Tissue engineering is a process that creates a three-dimensional (3D) porous scaffold that mimics natural tissues’ micro-environment, and in doing so, supports cell migration, adhesion, and proliferation to replace damaged tissues. Cells seeded into traditional tissue engineering scaffolds can only be attached to the surface of the scaffold, such that the distribution and migration of cells within the scaffold cannot be precisely controlled, with detrimental clinical effects. Over the last few decades, 3D printing has emerged as an additive manufacturing technology and been rapidly developed for the field of regenerative medicine. This technology overcomes the limitations of traditional cartilage tissue engineering by simultaneously constructing 3D artificial implants or complex biological tissues that combine with living cells, extracellular matrices, and other biological materials through a user-defined “bottom-up” model. Currently, 3D bioprinting technologies include inkjet bioprinting/droplet bioprinting, extrusion bioprinting, and laser-assisted bioprinting [1]. The primary advantage of 3D bioprinting in cartilage tissue engineering is that 3D bioprinted cells, hydrogels, and active substances can be distributed hierarchically and spatially according to the required 3D functions. The structure of interconnected pores and large surface area created by 3D bioprinting supports seed cell attachment, growth, inter-cell communication, and exchanges with gases and nutrients, which can significantly promote cartilage tissue regeneration over traditional solvent hydrogels.

In this review, we introduce various engineering methods that utilize 3D bioprinting equipment and describe several important types of hydrogels bioinks. We end with a discussion of the application of 3D printing hydrogel in the field of cartilage tissue engineering, and the future direction of 3D printing-based tissue engineering.

## 2. Bioprinting Technologies

3D bioprinting is an emerging technology that precisely dictates the construction of a three-dimensional living cell system in vitro through computer modeling. This process is a kind of rapid prototyping or additive manufacturing approach that uses layer-by-layer construction to build a tissue or organ. A digital model is first created using modeling software and then transmitted to the printer, which constructs the object by stacking layers of materials. The idea was first proposed by bioengineer Thomas Boland, the self-described “grandfather of bioprinting”, in 2000 [2]. After several years of development, this technology has overcome limitations of traditional tissue engineering technology.

To date, several 3D printing technologies have emerged. While inkjet printing was used at first, currently, micro-extrusion molding and light curing molding (e.g., fused deposition molding (FDM), selective laser sintering (SLS), optical mediated stereolithography (SLA), and digital light processing (DLP) are more commonly used (Figure 1). In the following sections, we will provide an overview of commonly used 3D printing technologies for the development of hydrogels, including detailed information on each manufacturing technology.

### 2.1. Inkjet-Based 3D Printing

Inkjet printing has been used to create 3D structures with multiple layers of droplets. Inkjet-based hydrogel 3D printing distributes very small volumes (1–100 picoliters) of low-viscosity bioink onto the substrate [3]. Of these approaches, a thermal-induction inkjet printer uses a thermal actuator to heat the liquid droplets, which rapidly expand to eject the bioink droplets from the print head. In piezoelectric-induction inkjet printers, the bioink is squeezed out of the chamber when pulses are applied using a piezoelectric actuator. Inject printing can successfully form the material into the required shape through the deposition process. It also has the advantage of controllable delivery of even very small volumes of solution to the nozzle. However, disadvantages of this method include the restriction that the biological materials used must form droplets in the form of liquid; in the end, the printed liquid must form a solid three-dimensional structure with detailed organizational structure and functions. Secondly, inkjet printers also have material viscosity limitations.

### 2.2. Micro-Extrusion Bioprinting

Micro-extrusion bioprinting uses a computer-controlled deposition system that uses pistons, pneumatic pumps, or screws to dispense hydrogel filaments onto the substrate through nozzles. The bioink used in extrusion bioprinting must have sufficient viscosity and cross-linking ability to maintain a three-dimensional structure during and after printing. Compared with other 3D-printing technologies, extrusion bioprinting can use a wider variety of materials, has a faster printing speed, and exhibits higher precision. A specific advantage over inkjet bioprinting is that extrusion printing technology can utilize a wider selection of bioinks. As such, extrusion printing is the most used method for cartilage bioprinting. Unfortunately, the printing accuracy is limited, the shearing force of the printing material on the nozzle wall can reduce the number of surviving cells, the printing process affects the cell viability, and the cells face dehydration and lack of nutrients after printing.

### 2.3. Laser-Based 3D Printing

Laser-assisted bioprinting relies on pulsed laser beams to generate pressure disturbances, which transfer the cell-containing material from the original printing material “ribbon” to the receiving substrate. Due to a lack of nozzle, laser-assisted bioprinting never experiences technical problems related to nozzles, such as nozzle clogging, and overcomes the problem of cell damage and death induced by shear stress generated when the nozzle diameter is extremely small and/or when the viscosity of the bioink is very high. A significant advantage, therefore, of this method is that it is compatible with the viscosity of a range of biological materials. Another key benefit of laser-assisted bioprinting is enhanced high spatial resolution (up to sub-micron resolution). This technology can also be adapted to print higher cell densities, e.g., to better control the interactions between cells and the high-definition mode of cells. However, due to expensive hardware and software, it is rarely applied to cartilage tissue engineering.

### 2.4. Stereolithography-Based 3D Bioprinting

Another emerging bioprinting method is stereolithography, which uses ultraviolet light to selectively crosslink bioinks in a layer-by-layer process. The precise movement of the ultraviolet light provides extreme control in the cross-linking of macromolecules and can stimulate the development of tissue structure. Key advantages of stereolithography are high resolution (<100 μm) printing and good cell viability (>85%). As with laser-based printing, this method lacks a nozzle and therefore avoids nozzle clogging. However, because they need high photopolymerization ability, few kinds of bioinks can be used. Furthermore, traditional stereolithography-based bioprinting requires the use of harmful ultraviolet light, which may cause cell mutations and cell damage. A new type of visible light cross-linkable bioink and cell adhesion has overcome this problem, and it greatly enhances the cell viability, which provides great potential in bioprinting and tissue engineering.

## 3. Bioinks for 3D Bioprinting

Bioink is the core component of 3D printing and typically contains a combination of scaffold materials, seed cells, growth factors, and various tissues and organs that need to be printed. Advanced bioinks use a variety of strategies to improve 3D printability and biocompatibility. For example, interpenetrating networks, nanocomposites, and supramolecular hydrogels exhibit shear thinning properties, which overcome the previous limitation with bioinks. Other emerging inks include functional groups and nanoparticles with biologically active properties that can greatly improve these biological functions [5] (Figure 2). At present, there are nearly hundreds of modified biological materials used as bioinks.

### 3.1. Scaffold Materials

Scaffold materials in bioinks are typically natural materials, including gelatin, alginate, collagen, silk fibroin, sodium hyaluronate, chitosan, and acellular extracellular matrix. Organic polymer materials, such as polylactic acid, polycaprolactone, polyethylene glycol, and polyglycolic acid, have been utilized. Inorganic materials are also sometimes added to scaffold material; these include nano-hydroxyapatite, tricalcium phosphate, graphene oxide, carbon nanotubes, nano-cellulose, iron oxide nano particles, and silver nanoparticles. However, cartilage has excellent mechanical properties due to its complex ultrastructure, which is difficult to replicate artificially. The use of nanotechnology can provide a solution in simulating the structure of cartilage tissue. Studies have proved that carbon nanotubes manufactured using 3D bioprinting technology can enhance the physical properties of cartilage scaffolds [6]. In another study, carboxylated cellulose nanocrystals (cCNCs) were prepared using ammonium persulfate as hydrogel inks, and stable cell-free and cell-loaded hydrogel inks with the best physicochemical properties and biocompatibility were developed [7]. We also used magnetic nanoparticles (Fe_2_O_3_) as a bioink to generation magnetic nanocomposite hydrogel for cartilage tissue engineering [8,9,10,11,12].

Hydrogels play an important role in 3D bioprinting [13]. Their excellent water absorption makes the hydrogel as the first choice for 3D applications. Nutrients and growth factors are encapsulated in the hydrophilic hydrogel to form a hydrogel network that mimics the microenvironment of natural tissues, allowing for high biocompatibility. Gelatin, alginate, hyaluronic acid, collagen, fibrin/fibrinogen, hyaluronic acid [14,15,16,17,18,19], chitosan [20], decellularized extracellular matrix (dECM) [21], and polyethylene glycol (PEG) [22] are commonly used bioinks, as they are natural materials with biocompatibility properties. Additionally, some of these materials can be easily photo-crosslinked in their modified form. Furthermore, there are some specific honeycomb integrins found within the hydrogel matrix that can enhance cell adhesion, migration, proliferation, and differentiation. At present, hydrogels are used in many clinical practice areas, such as spinal surgery and wound dressings [23] and for cartilage tissue engineering [24].

### 3.2. Cell Sourcing

Cell sourcing is one of the important parts of tissue engineering. It is an ongoing challenge to produce enough regenerative-non-immune cells that maintain their unique biological activity in the transplanted areas. For example, cartilage provides limited donor tissue for chondrocytes, while allogeneic or heterogeneous chondrocytes are often rejected due to the immune response of the human body. Various conditions, such as dedifferentiation and loss of cell phenotype, are prone to occur in the process of in vitro expansion. Current studies use progenitor cells or stem cells to overcome these issues, such as mesenchymal stem cells (MSCs) that can be isolated from various tissues and expanded and differentiated in vitro. Mesenchymal stem cells display strong self-renewal, proliferation, and differentiation potential. As MSCs can differentiate into chondrocytes under specific induction conditions [25,26], these cells are widely used seed cells in cartilage tissue engineering. MSCs can be sourced from bone marrow, adipose tissue [27], induced pluripotent stem cells [28], amniotic fluid [29], and synovial fluid [30,31]. However, stem cell-based 3D printing may result in enhanced tumorigenesis risk as well as genetic instability and chromosomal aberrations [32]. Bioinks prepared from articular cartilage progenitor cells (ACPCs), MSCs, and chondrocyte-loaded GelMA hydrogel outperformed chondrocytes in the production of new cartilage. Unlike MSC, ACPC also showed the lowest gene expression of type X hypertrophy marker collagen, and the highest expression of PRG4, which encodes an important protein (lubricin) in joint lubrication, suggesting that ACPC is a promising cell source for 3D bioprinting.

### 3.3. Growth Factors

Growth factors regulate the synthesis and metabolism of the chondrocyte matrix, promoting the differentiation of stem cells into cartilage and the proliferation of chondrocytes, as well as maintaining the phenotype of chondrocytes. As such, growth factors promote cartilage tissue regeneration and repair articular cartilage damage. Traditional growth factors used in tissue engineering include transcriptional growth factor β, insulin-like growth factor, bone morphogenetic protein, and fibroblast growth factor. For example, the transcriptional growth factor β family can effectively induce stem cells to differentiate into chondrocytes. In cartilage tissue engineering, transforming growth factor (TGF)-β1 and TGF-β3 are widely used to induce cartilage-derived differentiation of stem cells and maintain chondrocyte phenotype. Insulin-like growth factor (IGF)-I and fibroblast growth factor (FGF)-2 promote articular cartilage regeneration and protect adjacent joint tissues. In the bone morphogenetic proteins (BMP) family, BMP-2 and BMP-7 selectively induce mesenchymal stem cells to differentiate into chondrocytes. Fibroblast growth factor promotes the division and proliferation of fibroblasts and can also stimulate blood vessel formation, which plays an important role in wound healing and bone and cartilage damage repair.

New growth factors have been discovered in recent years, such as platelet concentrates [33] and small molecules, such as kartogenin (KGN) [34,35] and dexamethasone, that can regulate biological processes, promote the survival and proliferation of cells, and maintain the differentiated phenotype.

## 4. 3D Bioprinted Cartilage Tissues

Cartilage is a smooth tissue with a relatively low density of chondrocytes that lacks blood vessels and nerves, which can be found covering the ends of joint bones. We summary the 3D printing for cartilage tissue engineering in Table 1.

Although articular cartilage is only a few millimeters thick, it can prevent friction between joints and endure extreme load stress during limb movement. Cartilage defects, caused by aging, degenerative diseases, or trauma, commonly lead to the development of joint pain and arthritis. Despite many attempts, artificial cartilage cannot fully simulate the tissue composition, ECM, and mechanical properties of naturally occurring cartilage. Three-dimensional bioprinting can use a variety of materials and cells to create products in the desired shape, providing a huge opportunity for cartilage tissue engineering. Chondrocytes and MSCs are important seed cells for repairing cartilage damage. The scaffold design with gradient pore size and hole geometry can simulate the zonal organization-like structure of articular cartilage with similar mechanical strength characteristics to a certain extent. The 3D printed cartilage tissue gradient scaffold loaded with BMSCs will simulate the zonal like tissue of articular cartilage with higher cell viability, cell proliferation, type II collagen deposition, and cartilage gene expression. Thus, using a 3D scaffold, chondrogenic cells or MSCs can replace the damaged parts of cartilage tissue (Figure 3).

Hyaluronic acid (HA) is a naturally occurring non-sulfated glycosaminoglycan that plays an important role in synovial fluid and hyaline cartilage. Chondrocytes cultured using HA as a 3D scaffold showed higher expression levels of COL2A1 and proteoglycan (chondrocyte markers) [62]. Supplementing CS and HA alone or in combination can enhance the accumulation of glycosaminoglycan (GAG) and cell proliferation in the cartilage matrix embedded in the 3D fibrin-alginate hydrogel [63].

3D printed PCL stents have been widely used in cartilage engineering. However, the hydrophobic surface of PCL has poor cell affinity. Hybrid bioinks that incorporate PDA and PLGA nanoparticles into the PCL scaffold through 3D printing may significantly reduce the water contact angle of pure PCL and provide cells with a high biomimetic ECM microenvironment [54]. Biocompatible alginate hydrogels have proven their ability to create precise shapes of 3D printed structures. However, alginate is very soft and fragile, and the use of high-density collagen hydrogel improves the weight-bearing capacity of these joints. High-density collagen hydrogel produced printed materials have good mechanical stability and can support and maintain cell growth [64]. Printed hybrid bioink showed enhanced mechanical properties compared to the alginate or fibrin-collagen gel alone [43]. Schuurman et al. used an alginate-PCL hybrid material that could be used to encapsulate chondrocytes [65]. We found that gelatin/hydroxyapatite (HAP) hybrid materials assisted by enzymatic cross-linking enhance the gelatin scaffold’s ability to promote stem cell cartilage differentiation and can effectively repair the damage tissue [57]. Yang X et al. used 3D bioprinted cartilage tissue engineering by a collagen-alginate bio-ink, which can effectively maintain the phenotype of chondrocytes, and has excellent expansion rate and mechanical properties [60].

In addition, polycaprolactone (PCL) microfibers can be used to improve the mechanical properties of bioinks in the bioprinting process. Studies have demonstrated that extrusion-based bioprinting of alginate and agarose hydrogels supports the formation of hyaline cartilage more than the hydrogels of other groups. Mechanical studies that added PCL microfibers to the bioink could increase the elastic modulus of the bioinks, alginate, and GelMA by 544 times and 45 times, respectively [66].

A biomimetic multilayer osteochondral scaffold composed of PCL and HA/PCL microspheres through laser sintering (SLS) technology has been used for cartilage regeneration. SLS-derived scaffolds also demonstrate high biocompatibility and can induce articular cartilage formation in rabbit models of osteochondral defects [67]. Three-dimensional cell-printed Alginate/PCL scaffolds that contained TGF-β showed higher levels of ECM formation [36]. Kesti et al. developed a cartilage-specific bioink mixture of alginate and gellan extracellular matrix particles that were found to be superior to those of natural articular cartilage [37]. Markster et al. combined alginate and nanofibrillated cellulose, whose rapid crosslinking ability and shear thinning properties made stents manufactured by 3D printing more stable [51]. PEG/alginate hydrogel composites created by 3D printing exhibited higher fracture performance and higher cell viability than natural cartilage [68]. Daly et al. produced a composite for cartilage tissue engineering that included agarose, alginate, GelMA, and BioINK, and found that it was better than natural cartilage [39].

However, due to the difficulty of manufacturing scaffolds that fully mimic the microenvironment of natural cells, natural extracellular matrix (ECM) is very promising to ECM components for 3D printing in terms of providing possible ECM simulation capabilities for 3D printing structures. For example, Rathan et al., designed a new class of cartilage extracellular matrix (ECM) functionalized alginate bioink that supported the post-printing and cartilage formation of mesenchymal stem cells (MSC) and promoted the high-level expression of COLLII and ACAN in cells. When bioinks were loaded with MSCs and TGF-β3, they could support strong cartilage formation, making them suitable for direct “print and implant” cartilage repair strategies [41]. Other research has used ECM and silk fibroin mixed with bone marrow mesenchymal stem cells (BMSC) for 3D living cell bioprinting. This material enhanced the formation of cartilage BMSCs and optimized the cartilage repair environment, suggesting that this material may be an ideal scaffold for cartilage tissue engineering [42]. Using a synthetic PCL polymer with a gradient structure to sequentially print a hydrogel that releases two factors and MSCs, joint reconstruction and articular cartilage regeneration can be achieved [55].

In addition to natural biomaterials, there are synthetic hydrogel polymers that perform well as bioinks. For example, methacrylated poly (N-(2-hydroxypropyl) methacrylamide mono/dilactic acid) /PEG hydrogel containing 0.5% HAMA appears highly suitable for cartilage-like tissue regeneration [49]. Co-printing the hydrogel with PCL and HAMA can increase the stiffness of the composite scaffold to a value close to that of natural cartilage. In another study, thiol-functionalized HA was cross-linked with allyl-functionalized poly(glycidol) (P(AGE-co-G)) and used as a bioink to construct articular cartilage body tissue [50]. Compared with the bioink containing only PG, the bioink based on the combination of PG and HA showed improved cell viability and differentiation. Graphene oxide has also been used as a 3D scaffold material to support regenerated cartilage, providing a new way for the delivery of important growth factors. A 3D printed GO scaffold developed for the construction of cartilage matrices extended and matured along the boundary between the cartilage and the scaffold, significantly increasing collagen I expression in the cartilage [56].

Natural articular cartilage includes cells with different morphologies and arrangement, as well as various extracellular matrix (ECM) arrangements, compositions, and distributions. The structural heterogeneity and tensile properties of the tissue make it resistant to shear, stretch, and compressive forces exerted by the joints. By testing the distribution patterns of growth factors, mechanical gradients, and stem cells in each cartilage zone area, the bio-manufactured functional cartilage tissue can be improved to exhibit histological and mechanical characteristics.

## 5. Outlook

### 5.1. Advanced Developments in 3D Bioprinting Technology

3D bioprinting technology has unique advantages in the field of preparing high-precision and controllable renewable stents. So far, a variety of 3D printing technologies, reviewed above, have been used to manufacture different tissues and repair damaged stents. Among them, FDM printing can prepare scaffolds with high porosity and strong mechanical properties. However, due to printing conditions, this method cannot print with cells or growth factors, and therefore cannot meet the demand for cartilage regeneration in the repair of osteochondral defects. DLP printing technology, based on photopolymerization, can flexibly print hydrogel inks loaded with living cells or biomolecules. Therefore, a combination of DLP and FDM printing technologies that can prepare double-layer scaffolds holds great significance in the research and application of cartilage defect repair and regeneration. This combination has been used to prepare a GelMA hydrogel with interleukin 4 (IL-4) on the upper layer and porous polymer on the lower layer, which demonstrated excellent anti-inflammatory activity in vivo and in vitro and significantly enhanced cartilage repair and subchondral bone regeneration.

Moving forward, 4D printing is a conceptual technique that could use “smart” 3D structures that can be programmed to change their shape and function in response to external stimuli, such as heat, ultraviolet light, current, or pressure. Interestingly, the manufacturing of bioprinting functional tissues that do not require support has recently attracted attention. For instance, researchers have used freshly printed human placentas to study the transportation of nutrients from mother to fetus. This method can be used to understand various life-threatening situations that may occur during pregnancy and childbirth. However, much research is still needed to successfully manufacture compatible tissue transplants and whole organ transplants. In general, we believe that, in the near future, 3D bioprinting will reach new heights through patient-driven precision medicine and complex tissue manufacturing.

### 5.2. 3D Bioprinting for Cartilage Tissue Engineering

3D bioprinting of scaffolds with living cells for potential cartilage tissue engineering is very interesting. Therefore, the development of bioinks is very important for cell bioprinting. To quickly create accurate 3D structures, it is necessary to develop new bioink materials and to improve the accuracy of current bio-printing equipment. Bioinks are described as “cell preparations suitable for processing by automated bio-manufacturing techniques”. Bioinks are hydrogel preparations containing single cell suspension or cell aggregates. They can also be used in combination with cell-free bio-material inks, such as exosomes. Three-dimensional printing has been well applied in cartilage tissue repair.

The 3D scaffold can be used as a platform for the release of exosomes in the joint tissue repair area [69]. Exosome-laden 3D materials can be loaded with small molecule compounds, such as miRNAs that are used for drug delivery, and used as cell-free therapeutic products for tissue repair [70,71]. Additionally, the engineering exosomes that are printed by 3D printers can realize the targeted drug delivery for cartilage repair [72,73].

Although bioprinting typically involves repeatedly depositing bioinks onto a surface, such that the 3D structure is built by the layering of printing filaments, there is currently an emerging technology that deposits bioinks into a suspension during the printing process.

There are many opportunities for bioprinting to solve basic biological problems outside traditional medicine. For instance, extrusion bioprinting can span the cell matrix, cell soluble factors, and cell–cell interactions that drive biology to form a variety of cell morphologies [74]. This can be achieved by selecting bioinks that control the local cellular microenvironment (i.e., manipulate biochemical and biophysical signals), and by placing the printed bioinks to influence the macro structure and cell population dynamics.

The commercialization of bioprinting has accelerated the development of this field. However, challenges remain in the design of suitable bioinks and complex tissue manufacturing. Maintaining the viability of the cells encapsulated in the bioink and ensuring that they are not damaged during the printing process requires new bioink formulations, new cell sources, and advanced printing technology.

The biggest challenge remaining is to develop a biomimetic cartilage structure that can simulate the gradient and the signal transduction mechanism in different layers to induce region-dependent cartilage directional differentiation and ECM deposition. Previous studies have demonstrated that a scaffold with a smaller pore size (100~200 μm) better promotes cartilage formation in osteochondral regeneration [75]. However, pores this small are too tiny to inhibit bone formation and angiogenesis, curtailing the delivery of oxygen and nutrients and the integration with host tissues. Other studies have applied hydrogels to the problem of cartilage regeneration. However, as hydrogels exhibit poor mechanical properties and printing performance, it is still difficult to construct large-scale cartilage tissues using this method. Future 3D printing technologies that construct MSC-loaded cartilage tissue with dual-factor release and gradient structure may increase the effectiveness of cartilage regeneration and repair in vivo.

### 5.3. Challenges and Limitations of 3D Bioprinting in Clinical Transplantation Applications

A major challenge in 3D bioprinting technology is the limited use of bioinks, because they must have unique and optimized properties in order to be used in clinical applications. These properties include insolubility in vivo and in culture, structural stability, tissue degradation consistent with regeneration, promotion of cell growth, and non-toxic properties. Bioinks must also integrate with other cells and allow vascularization. At the same time, bioinks can be negatively affected by the bio-printing process, reducing cell viability. Currently, no bioink materials meet all these requirements.

We are also yet to achieve in situ bioprinting, in which living tissue can be printed directly to the defect in the operating room. Challenges in this process include maintaining a sterile surgical field while also including a well-integrated printer and surgical procedure. In addition, there are ethical considerations, as this process requires a multidisciplinary approach to disclose sensitive medical information to doctors, engineers, and others involved. This process must also comply with regulatory standards for clinical use and should be affordable before it can become commonplace.

To date, various 3D bioprinting technologies have been used to study tissue engineering applications aimed at simulating various tissues and organs. Three-dimensional bioprinting paves the way for the integration of biomaterials, imaging, modeling, and computing technologies in the fields of biomedicine and tissue engineering. Three-dimensional bioprinting technology can adjust the shape, porosity, and size of 3D scaffolds, with key application in research and clinical settings. There remain challenges to the wide-spread adoption of these techniques, including the development of biological inks for bioprinting tissues or organs. Traditional 3D bioprinting focuses on the creation of cell-free 3D structures, while recent 3D bioprinting technology uses cells to generate 3D bioactive structures quickly and accurately in one step. The future multidisciplinary cooperation between biologists, bioengineers, and doctors will provide broad prospects for the application of 3D bioprinting in cartilage regenerative medicine.

## Figures and Tables

**Figure 1 gels-07-00144-f001:**
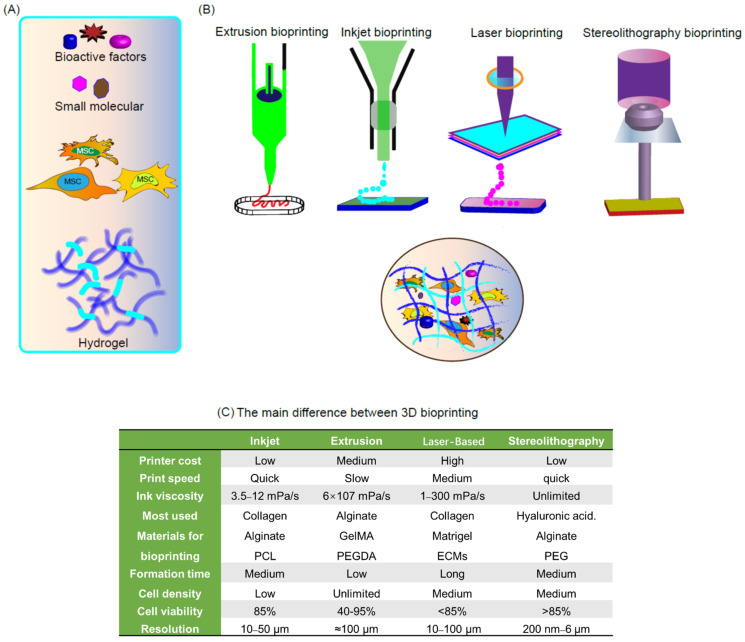
Schematic diagram of bioprinting technology and comparison of bioprinter types. (**A**) bioink preparation for 3D bioprinting; (**B**) schematic representation of the 3D bioprinting technologies—inkjet bioprinting, extrusion-based bioprinting, laser-assisted bioprinting and stereolithography-based 3D bioprinting; (**C**) difference between the types of 3D bioprinting. Reproduced, with permission, from [4].

**Figure 2 gels-07-00144-f002:**
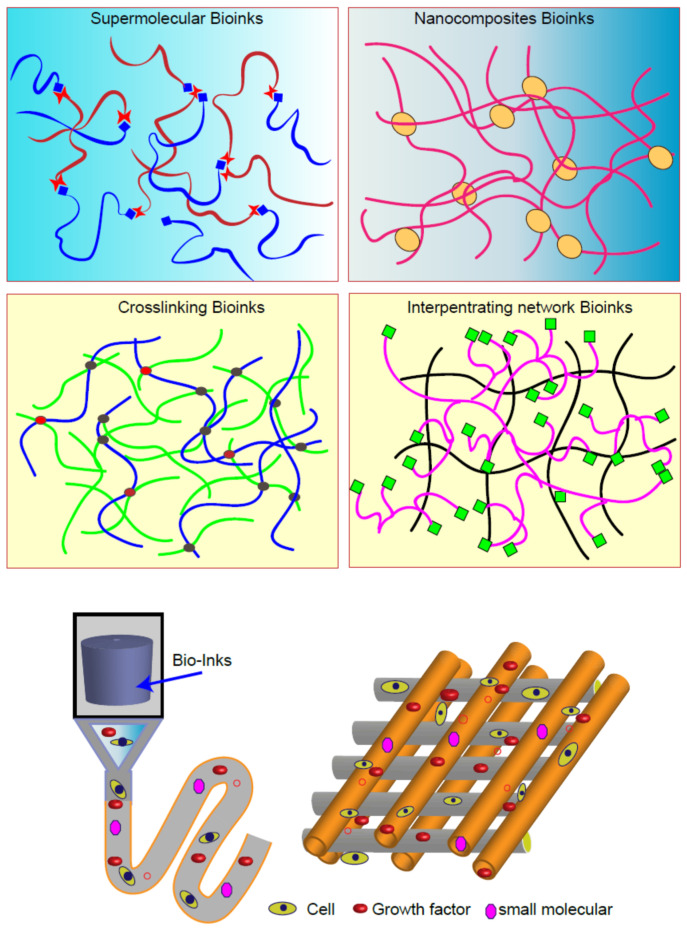
Schematic summary of various bioinks for 3D bioprinting of biomaterials.

**Figure 3 gels-07-00144-f003:**
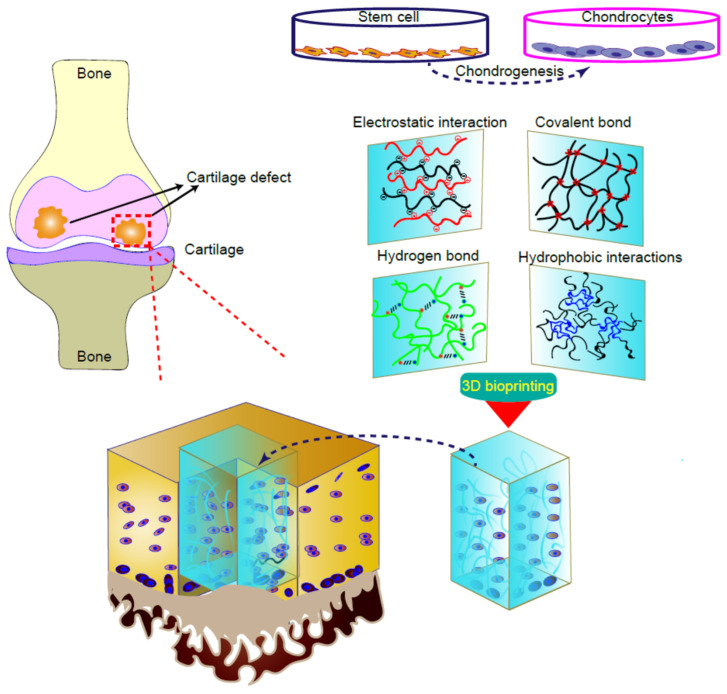
The procedure of creating artificial cartilage tissues with biological functions via 3D bioprinting. The regional characteristics of natural cartilage and the current 3D printing methods used for cartilage defect repair. With the 3D printing technology, the structure is captured from different areas of the cartilage, and the seed collagen fibers are formed into a layered scaffold; they are distributed in ribbons to simulate the structure of natural cartilage.

**Table 1 gels-07-00144-t001:** Summary of various methods and materials for 3D printing for cartilage tissue engineering.

Biomateria	Bioprinting Method	Cell Type	Function	Ref.
Polycaprolactone/Alginate	Extrusion-based	Chondrocyte	Cartilage regeneration	[36]
Polysaccharides/gellan/alginate/BioCartilage	Extrusion-based	Chondrocyte	Supports proliferation of chondrocytes	[37]
Collagen type II hydrogel	Extrusion-based	Chondrocyte	Promote biomimetic chondrocyte density gradient and formation of type II collagen hydrogel structure	[38]
Gelatin	Extrusion-based	MSCs	Maintain MSC viability (~80%), development of hyaline-like and fibrocartilage-like tissue	[39]
Gelatin	Extrusion-based	Chondrocytes, chondroprogenitor cells, MSCs	Supports the synthesis of new cartilage in stratified co-culture	[40]
Cartilage extracellular matrix (cECM)/alginate	Extrusion-based	MSCs	Prmote COL 2 and ACAN expression and cartilage formation	[41]
ECM/silk fibroin	Extrusion-based	BMSCs	Supports the controlled release of cartilage growth factors and enhances the formation of cartilage	[42]
Fibrinogen/fibrin	Inkjet printing	Rabbit chondrocytes	Enhance mechanical properties and cartilage ECM	[43]
PCL/GelMA	Inkjet printing	MSCs and chondrocytes	Native-like collagen anisotropies	[44]
PCL-alginate	Extrusion	Chondrocytes	Promotes the formation of cartilage tissue and type II collagen	[45]
Polysaccharides, Gellan, Alginate	Extrusion	Chondrocytes	Enhances deposition of cartilage matrix proteins	[37]
Norbornene-modified hyaluronic acid (NorHA)	Extrusion	MSCs	Induces chondrogenesis and cartilage formation	[46]
Nanocellulose/Alginate	Extrusion	Induced pluripotent stem cells (iPSCs)	Induces chondrogenic and cartilage production	[47]
Gelatin, PLGA	Extrusion	Chondrocytes	Promotes cartilage regeneration and maintenance of cartilage tissue shape in vivo	[48]
HAMA/pHPMA-lac/PEG	Extrusion-based	Equine chondrocytes	Promotes cartilage-like tissue formation	[49]
PG-HA, allyl-functional PGs	Extrusion-based	Human and equine MSCs	Promotes chondrogenic differentiation	[50]
Nanocellulose-Alginate	Extrusion-based	Chondrocytes	Maintain cell viability of 73% to 86%	[51]
Polyurethane, Hyaluronic Acid	Extrusion-based	Wharton’s jelly mesenchymal stem cells	High cytocompatibility	[52]
PLA/Alginate hydrogel	Extrusion-based	Human adipose-derived stem cells	Exhibits high levels of cell proliferation; promotes ECM secretion and chondrogenic differentiation	[53]
PLGA/PDA/PCL	Fused Deposition Modeling	Chondrocytes and rBMSCs	Continuous IGF-1 release and better cartilage formation ability	[54]
PLGA/Hydrogel/PCL	Extrusion-based	BMSCs	Dual-factor releasing and gradient-structured	[55]
Graphene oxide (GO)/chitosan/collagen type-I	Inkjet printing	Chondrocyte	Cartilage-matrix regeneration	[56]
Gelatin/hydroxyapatite	Microextrusion	hUCB-MSCs	Articular cartilage repairs	[57]
Silk/Gelatin	Extrusion-based	Chondrocytes	Biocompatibility	[58]
Hydroxybutyl chitosan/oxidized chondroitin sulfate	Inkjet	Human adipose-derived stem cells	Multifunctional cell delivery hydrogels for the cartilage repair	[59]
Collagen/alginate	Extrusion-based	Chondrocytes	Inhibit chondrocytes dedifferentiation and miantain the phenotype	[60]
Silk/Fibroin/Gelatin	Extrusion-based	BMSCs	Inhibited the dedifferentiation of chondrocytes and maintained the phenotype.	[61]

## Data Availability

The data that support the findings of this work are available upon reasonable request from the authors.

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
