# Peer review of "3D Bioprinting of Hydrogels for Cartilage Tissue Engineering"

_gels, 2021, doi:10.3390/gels7030144_

Round 1

Reviewer 1 Report

3D bioprinting that could provide scaffolds with ideal mechanical properties and biological functions are of great interest for cartilage tissue engineering. In this review paper, the authors first overviewed the current bioprinting techniques and compared the advantages and limitations for four most used manufactory technology. Then they summarized the formulas of bio-inks composed of various polymer scaffolds as well as cell sources, and discussed their applications for cartilage tissues. Finally, the authors revealed the current challenges in the field and pointed out some future directions for the development of 3D bioprinting. This manuscript presented a topic with broad interest for materials science and tissue engineering. However, several aspects of the manuscript should be improved prior to consideration for publication.

Comments that should be addressed:

1) In Fig 1c, the authors included a table to compare four 3D bioprinting techniques, which is important to summarize their pros and cons. However, some of the numbers in this table didn’t match the ones discussed in the manuscript. For examples, on Page 3 Line 120, the authors mentioned that the cell viability is > 90%. This number changed to > 85% in Fig. 3c. The authors should make these numbers consistent. Also, the ‘inkjet’ method was likely misspelled.

2) Following the last comment, in Fig 1c, it would be helpful to add the quantitative number of resolutions instead of qualitative evaluation (high or low).

3) In section 2.4, the authors stated that stereolithography-based 3D bioprinting could provide high cell viability. However, at the end of this paragraph, the authors also commented that the limitation of this technique is the damage of cells and the risk of genetic DNA mutations. Are these comments conflicting with each other?

4) On Page 4 Line 142-145, the authors summarized a series of natural polymers used for hydrogels. PEG was mentioned in this listed, but it is a synthetic polymer, not a natural polymer.

5) In Fig. 2, the authors showed the schemes of various bio-inks. And in Fig. 3, the authors described the procedure to implant scaffold biomaterials for cartilage tissues by 3D bioprinting. However, Fig.2 and Fig.3 were not discussed and rarely mentioned in the manuscript. The authors should either remove these two figures or include some paragraphs of discussion related to these figures.

6) In section 4, the authors summarized previous studies using 3D bio-printing for cartilage tissues. As the main focus of this review paper, this section needs to be significantly improved. First, the organization of this section is not clear to readers. The authors listed several studies that were unrelated with each other and put them together in the same paragraph (eg: second paragraph on Page 7). Ideally, the authors could discuss related studies in the same paragraph, such as using the same biomaterials, printing techniques or therapeutic cell sources, and summarize the status at the end of paragraph. Second, only ~ 15 papers were discussed in the section and listed in Table 1. However, many papers were not included in this review including some highly cited ones (eg: 3D bioprinting via an in situ crosslinking technique towards engineering cartilage tissue Sci Rep 2019, 9, 19987; Cartilage Tissue Engineering by the 3D Bioprinting of iPS Cells in a Nanocellulose/Alginate Bioink Sci Rep 2017, 7, 658; 3D Bioprinting of Spatially Heterogeneous Collagen Constructs for Cartilage Tissue Engineering ACS Biomater. Sci. Eng. 2016, 2, 10, 1800; etc). Overall, more citations are required for a review paper.

7) In section 5, the authors provide an insight into the challenges and opportunities of 3D bioprinting for cartilage tissue, which is important and valuable for the future direction of this field. However, the authors focused solely on 3D bio-printing technique in section 5.3 – 5.5, which has been previously discussed a lot in review articles. Instead, the authors should focus more on perspective of 3D bio-printing specifically for cartilage tissue applications.

8) In section 5, citations were missing when the authors refer to previous studies. For example, on Page 10 Line 304, the authors commented that “Previous studies have shown that ….” The authors should cite related papers to support their discussions.

9) Some acronyms of the technical terms were used before mentioning their real name in the manuscript, including MSC, TGF and IGF, etc. The authors should provide the first use of an abbreviation immediately after the expanded form.

10) The language needs to be improved in the manuscript to avoid some grammar mistakes. For example, on Page 4 Line 136, the authors listed the scaffold materials by saying that “These scaffold materials include natural materials: gelatin… extracellular matrix, etc.; Organic polymer materials include … polyglycolic acid, etc.; commonly used inorganic materials include … silver nanoparticles, etc.” Natural materials, organic polymer materials and inorganic materials all belong to scaffold materials. However, in this discussion, it sounds like scaffold materials, organic polymer materials and inorganic materials were discussed in parallel. On Page 5 Line 151, “…. which can enhance cell adhesion, migration, proliferation, and differentiation of the.” This is not a complete sentence. On Page 5 Line 170-173, this following sentence has two verbs “In addition to bone marrow mesenchymal stem cells, there are currently adipose-derived mesenchymal stem cells….. synovial fluid mesenchymal stem cells are cell sourcing that are widely used in cartilage tissue engineering”. On Page 5 Line 176, the author stated that “the long-term in the vitro culture of MSC….”, which should be in vitro culture. On Page 5 Line 196, “In addition, the development of small molecules can regulate biological processes”, the development is not supposed to regulate the biological processes.

Reviewer 2 Report

The manuscript presents in a synthetic manner the main components of 3D bioprinting for the construction of cartilage tissue. 

Reviewer 3 Report

In this manuscript (gels-1361726), the authors have reviewed the progress on 3D bioprinting of hydrogels for cartilage tissue engineering applications. This review could be of potential interest for researchers, but authors have mainly focused to write on typical information available and discussed in most of the review articles, and only collected surficial literature on this particular topic by writing one and a half pages for 3D bioprinted cartilage tissues (section 4). There are many articles on 3D printing of materials for cartilage tissue engineering applications, which were not considered by the authors. For example, Tissue Engineering Part C: Methods, 22(3), 2016, 173-188; Acta biomaterialia, 121, 2021, 193-203; Journal of Nanomaterials, 2020, 2020, Article ID 2057097; Scientific reports, 7(1), 2017, 1-10; ACS Biomaterials Science & Engineering, 2(10), 2016, 1800-1805; Biomedical Materials, 14(2), 2019, 025006; ACS applied materials & interfaces, 11(37), 2019, 33684-33696; ACS biomaterials science & engineering, 3(11), 2017, 2657-2668; Materials, 14(14), 2021, 3977; etc.

Authors have only considered overall 45 references and only15 references randomly by putting together for the article. After going through the manuscript, I have found that this review article does not provide any new research directions for future studies. Therefore, I would not recommend it for consideration in the present form.  

Round 2

Reviewer 1 Report

The authors have nicely addressed most of my comments. However, several aspects of the manuscript still need to be improved.

Comments that should be addressed:

1) In Fig. 3, the authors described the procedure to implant scaffold biomaterials for cartilage tissues by 3D bioprinting. Although the authors added 1 or 2 more sentences in the discussion, it is not comprehensive. For examples, in Fig. 3, chondrogenesis and multiple physical interactions were described. However, the authors didn’t comment anything in detail in the first paragraph in section 4. The authors should provide more detailed discussions about each figure since there are only 3 in this review.

2) In section 4 about 3D bio-printing for cartilage tissues, they authors have added more papers and improved the organization in the discussion. However, the organization of this section is still unclear. Here are the explanations:

The main idea of each paragraph (2- 7) in section 4 here is: examples, PCL microfibers, hydrid bioinks, HA scaffold, high bear-loading scaffold, synthetic scaffold. Paragraph 2 has no key idea but only random examples. And there are many overlaps between these paragraphs. In paragraph 4, the authors started with “hybrid bioink scaffold”. However, in the examples in paragraph 2 about HA/PCL bioink, in paragraph 3 about incorporating PCL fibers and in paragraph 6 about high bear-loading environment, all of them could also be considered as hydrid bioinks. Also, in paragraph 6 about the challenges to provide high bear-loading environment, it seems that none of these examples mentioned mechanical resistance, which is irrelevant with the key concept in this paragraph. All of these are confusing to readers. They authors should organize these examples in a better way.

Moreover, (YEAR) appeared in the manuscript many times. The authors should either remove all these or put the exact year in the parenthesis.

Reviewer 3 Report

Authors have improved the manuscript in some way, but there is still a window to improve further to make this paper effective for publication and wide readership.

Because various nanoparticles have also been used in preparing bio-inks, for example, nanocellulose, nano-hydroxyapatite, carbon nanotubes, etc., authors should also introduce them a little before staring the description the sentence using them into hydrogel-inks by various studies (descripted in this manuscript). For example, Biofabrication12(2), 2020, 025029; Frontiers in Materials6, 2019, 313; Materials13(18), 2020, 4039; etc.

In addition, authors should also include comprehensive literature for 3D printed biomaterials for cartilage applications, which are still not included, for example, Advanced materials, 2017, 29(29), 1701089; Materials Science and Engineering: C83, 2018, 195-201.

After this revision, this manuscript can be accepted for publication

Round 3

Reviewer 1 Report

The authors have addressed my comments. This manuscript is acceptable now.